# Rural Knowledge Transformation in Terms of Mercury Used in Artisanal Small-Scale Gold Mining (ASGM)—A Case Study in Gorontalo, Indonesia

**DOI:** 10.3390/ijerph20176640

**Published:** 2023-08-24

**Authors:** Andi Patiware Metaragakusuma, Masayuki Sakakibara, Yayu Indriati Arifin, Sri Manovita Pateda, Mohamad Jahja

**Affiliations:** 1Research Institute for Humanity and Nature, Kyoto 603-8047, Japan; sakaki@chikyu.ac.jp; 2Department of Earth Science, Graduate School of Science and Engineering, Ehime University, Matsuyama 790-8577, Japan; 3Faculty of Collaborative Regional Innovation, Ehime University, Matsuyama 790-8577, Japan; 4Department of Geology, Universitas Negeri Gorontalo, Gorontalo 96100, Indonesia; yayu_arifin@ung.ac.id; 5Faculty of Medicine, Universitas Negeri Gorontalo, Gorontalo 96100, Indonesia; manovitapateda@ung.ac.id; 6Department of Physics, Universitas Negeri Gorontalo, Gorontalo 96100, Indonesia; mj@ung.ac.id

**Keywords:** rural knowledge, mercury, artisanal small-scale gold mining, ASGM, TDCOPs, TBO

## Abstract

Gorontalo is reputed to be one of the best-quality gold producers in the Indonesian archipelago. Gold production has been largely achieved through the activities of artisanal small-scale gold mining (ASGM), which as part of its extraction process, primarily uses mercury—a substance known to cause negative impacts on health and the environment, leading also to numerous socio-economic issues. This research aims (1) to investigate the extent of rural knowledge regarding mercury and to determine whether a video that explains mercury and the problems that occur as a result of ASGM can significantly transform rural populations’ knowledge; (2) to inspect different factors separating the SR group (those who live in the same regency as the ASGM area) from the NR group (those who live in the neighboring regency/city of the ASGM area) and to find out whether said factors are statistically significant. The results show that the test subjects’ knowledge of mercury had increased after watching the video, and that their willingness to oppose ASGM activities is one of the significant factors within the two groups. Moreover, this paper briefly describes the follow-up activities of the SRIREP project (Co-creation of Sustainable Regional Innovation for Reducing Risk of High-impact Environmental Pollution) in encouraging rural communities to explore sustainable livelihoods as an alternative to ASGM.

## 1. Introduction

The Ministry of Environment and Forestry (KLHK) of the Republic of Indonesia defines artisanal small-scale gold mining (ASGM) as small-scale gold mining activities that are managed by either an individual or small groups/enterprises which generally operate informally without permission/a license and exploit marginal gold reserves with limited capital investment and production. ASGM areas are mostly located in remote rural areas with hard-to-reach access, such as protected forests and conservation areas. In some places, ASGM activities are conducted in residential areas [1]. Based on data from the World Gold Council, Indonesia ranks as the ninth-highest gold producer in the world, with a pro-duction value of nearly 120 tons of gold in 2021 (data as of December 2021) [2], most of which are being supplied from large mines spread across various provinces [3]. However, the country is also host to a large number of ASGM activities [1]. ASGM is operated by the rural community, who generally believe, by custom, that land rights holders can carry out mining. As such, many of them consider it legitimate to depend on mining for their livelihood. In contrast to this, governmental law states that mining can only be performed if one is in possession of a mining permit. This fundamental disagreement is what has led to the emergence of much ASGM.

In many countries, including Indonesia, ASGM primarily uses the traditional mercury amalgam method for extracting gold; this is because this method is faster, requires less effort, and is more affordable compared to other existing methods [4,5]. The work involved in such mining activities is considered by many people as accessible to anyone as it requires no special training, so most rural communities choose to involve themselves fully or partially in mining activities. As such, it is not surprising that the number of artisanal small-scall gold mines, including unlicensed ones, continues to grow, with more than 2600 ASGM sites spread throughout Indonesia [6]. Figure 1a,b illustrate the condition of one of ASGMs in Indonesia, located in Suwawa Timur subdistrict, Gorontalo. Furthermore, the existence of ASGM is considered to also have an array of economic impacts on the development of micro and small businesses. Many new businesses have emerged at the mining sites, such as transportation and goods delivery services (Figure 1c), food stalls (Figure 1d), small grocery stores (Figure 1e), small laundries that are run by women (Figure 1f), etc., to meet the miners’ needs.

It is recorded that at present, there are approximately two million people whose livelihoods depend on ASGM activities in Indonesia, the majority of whom being locals who have been doing such work for generations, as well as those who come from elsewhere seasonally [7].

However, ASGM is widely known as the biggest contributor to (at least 38%), and largest emitter of global mercury pollution in the atmosphere [8]. ASGM not only negatively impacts the environment, but also impacts the health of miners, their families, and the surrounding communities [9,10]. The mercury vapor released from the amalgam-burning process is extensively diffused, leading to mercury infiltration into the bodies of miners and other inhabitants through inhalation [11]. Mercury poses an especially serious threat during human fetal and early childhood development. It can be toxic to the nervous, digestive, and immune systems, as well as skin and internal organs [5].

In Indonesia, mercury is classified as a hazardous and toxic material (B3), whose use is prohibited by law [12,13]. The government of Indonesia has played an active role in mercury control since the establishment of the Intergovernmental Negotiating Committee (INC) on the Legally Binding Instrument of Mercury, organized by the United Nations Environment Programme (UNEP) from INC-1 in 2010 to INC-5 in 2013 in Geneva. The government of Indonesia then signed the Minamata convention in October 2013 in Kumamoto, Japan, which was then officially ratified in 2017 by domestic law number 11 of 2017 in September 2017 [14]. Two years later, in April 2019, Presidential Regulation No. 21/2019 on the National Action Plan for Mercury Reduction and Elimination (RAN-PPM) was enacted as a National Priority Program by the Indonesian government, which aims (1) to ban the use of mercury in ASGM and (2) to reduce and eliminate mercury contamination. In September 2020, the government of Indonesia then proposed the Regulation of the Minister of Energy and Mineral Resources of the Republic of Indonesia Number 16 of 2020 concerning the Ministry of Energy and Mineral Resources Strategic Plan 2020–2024 [15] for Mercury Reduction and Elimination by 2030. Based on the RAN-PPM report, efforts to reduce mercury use in the ASGM sector resulted in a reduction of 10.45 tons of mercury. This was achieved through curbing the ASGM sector that uses mercury and through efforts to develop gold processing methods without mercury [16]. The government continues to crack down on its use in illegal mining, and various measures have been taken to crack down on its employment in illegal mining.

Since there have not been many studies investigating the extent to which communities are aware of the hazards of mercury used in gold processing in artisanal mines in Indonesia, it is assumed that the impacts of mercury may not be widely recognized in the community, both by those who live near ASGM sites and those who do not. Thus, in this study, the extent of public knowledge regarding mercury and its dangers will be investigated, and as part of this, it will be determined as to whether the presentation of a video which explains mercury and the problems that occur as a result of ASGM can significantly transform the public’s perception of mercury. In addition, this study will inspect different factors between the SR group (those who live in the same regency as the ASGM area) and the NR group (those who live in the neighboring regency/city of the ASGM area) and assess whether these data are statistically significant.

Furthermore, the activities conducted by the SRIREP project (Co-creation of Sustainable Regional Innovation for Reducing Risk of High-impact Environmental Pollution), together with multisectoral stakeholders, in developing regional potency for the creation of alternative sustainable livelihoods through a transdisciplinary research approach as a follow-up action to rural knowledge changes with respect to mercury will also briefly be described in this paper. 

## 2. ASGM in Gorontalo

Gorontalo was one of the regencies in North Sulawesi province with the largest ethnic group population among other regencies, and in 2000, it was officially recognized as Indonesia’s 32nd province. Administratively, Gorontalo province consists of five regencies and one city [17]. Based on the result of the 2020 Population Census (conducted every 10 years), Gorontalo province’s population had reached a total of 1.17 million people (less than 1% of the Indonesian population), which ranks third place as one of the lowest populated areas in Indonesia [18]. Gorontalo is included in the six provinces with the smallest total area in Indonesia [19]. 

Gorontalo has abundant natural resources covering five sectors including the mining sector [18]. According to the records of Johann Friedrich Riedel, a German missionary who was sent by the Nederlandsch Zendeling Genootschap (NZG) in 1831, almost all the land in Gorontalo contains gold minerals [20]. The locations of Gorontalo’s gold potential can be found in subdistricts in North Gorontalo Regency, Pohuwato Regency, Boalemo Regency, Gorontalo Regency, and Bone Bolango Regency [17,21,22,23,24,25] (see Table 1). Gold mining in Gorontalo began in the Sumalata region in 1831 [26,27] then spread to the Pohuwato Region in the 1970s [28] before expanding to the Bone Bolango Region in the 1990s [21,29]. With this huge gold potential, it is unsurprising that ASGM activities have been taking place in Gorontalo. Since the kingdom era in the 18th century, Gorontalo has been known as one of the best-quality gold producers in the Indonesian archipelago [20]. The distribution of potential gold in Gorontalo is shown in Figure 2.

At present, there are two mining companies in Gorontalo Province that carry out contracts of work from the government (also known as “kontrak karya” in the Indonesia language) for the production of gold commodities and other resulting minerals. The first is PT Gorontalo Minerals (GM) in Bone Bolango Regency, which is located around Mak River, with estimated gold reserves of 3.6 million troy oz. Additionally, there is also PT Gorontalo Sejahtera Mining (GSM) in Pohuwato Regency, which is located in the Pani block, with estimated gold reserves of 2.2 million troy oz [31], as well as eight companies holding gold and copper mining licenses, with the remainder of activities being carried out via ASGM operations that employ thousands of traditional gold miners [21,31].

Based on data from the Department of Mining and Energy of Gorontalo Province, of the many community mining areas (WPR) where ASGM in conducted in Gorontalo Province, only one WPR has obtained a mining license (IPR), namely, the WPR in Bumela village, Bilato sub-district. The rest have not obtained a permit yet and have been designated the status of unlicensed artisanal small-scale gold mines (also known as PETIs). Such mines operate without being associated with companies that already hold licenses.

Bone Bolango Regency has the largest estimated gold reserves, which are centered around the Suwawa area. An inevitable consequence of this is that the area is host to a lot of ASGM sites. Mining business areas of gold in Bone Bolango Regency can be seen in Figure 3.

Based on several research results related to mercury in Gorontalo, it was found that the mercury content in the hair of residents in ASGM locations was higher than that of non-ASGM residents, and that the mercury content exceeds the WHO/IPCS threshold in fish and sediment [21,32,33]. Recent studies have found that the mercury content in the soil affects the distribution of Pteris vittata plants, correlating to a visible decrease in the number of these plants around the ASGM area [34].

Based on data from the Gorontalo Provincial Environment and Environment Office, there are more than ten thousand miners with 211 trommels spread across the ASGM sites in Gorontalo [35]. In general, miners engaged in ASGM exploit underground hard rock deposits, with only a few exploiting alluvial deposits. The deposits are then extracted, commonly using a trommel and the addition of mercury, to free the gold from other minerals. Mercury serves to capture the gold, resulting in amalgam (an alloy of gold and mercury). To obtain pure gold, a combustion process is required to vaporize the mercury, followed by a refining process of further heating to remove the mercury and other impurities [8,36,37,38].

## 3. Material and Methods

### 3.1. Study Site

This study was undertaken based on purposive sampling in 4 sites (marked by red dots in Figure 4); 2 sites are from the same regency as ASGM, i.e., Bone Bolango Regency, and the 2 other sites are from the neighboring regency/city of ASGM sites: 1 site from Gorontalo City, and 1 site from Gorontalo Regency. The involvement of the respondents from outside of ASGM areas ensures that fairness and credible information is obtained from both groups of people. Furthermore, two of the three neighboring regions that border Bone Bolango Regency were used as sample sites, ensuring a fair and credible distribution of sampling locations. In addition, Gorontalo City was designated as one of the sample sites because it is a regional neighbor of Bone Bolango Regency (which has the most ASGM sites, see Table 1). Opportunities for residents of Gorontalo City, especially those living in the suburbs bordering Bone Bolango, are considered to be high [29], so it was deemed a necessity to involve the city’s inhabitants, especially those who are living near the city border.

### 3.2. Data Source and Analyzing

For the purposes of this research, data were collected using both quantitative and qualitative approaches. For the quantitative data, a cross-sectional approach where data were collected from respondents at a single point in time was employed using 2 types of questionnaires: type 1 being filled out before watching the video, and type 2 being filled out after watching the video. The video screening was carried out immediately after conducting the first survey “before watching the video”, and the survey “after watching the video” was conducted immediately after watching the video. For the qualitative data, site observation, and in-depth interviews were used to draw and complete the study’s conclusions.

A total of 210 valid questionnaires, consisting of questionnaires before (105) and after watching (105) the video, were collected in July 2022. The obtained data were analyzed via Statistical Packages for the Social Sciences (SPSS): (1) paired samples *t*-test to determine whether the presented video can significantly transform rural knowledge in terms of mercury; (2) independent samples *t*-test to inspect factors that showed statistically significant differences within 2 groups. Respondents were divided into two groups: (a) those who live in the same regency as ASGM areas—categorized as the “SR” group with a total of 64 respondents; (b) those who live in the neighboring regency/city of ASGM areas—categorized as the “NR” group with a total of 41 respondents. The normality test for ensuring the variable is normally distributed was performed first before entering paired samples *t*-test and independent samples *t*-test. The parameters considered were a confidence level of 95% where the research error rate is not greater than 5%, indicated by the *p*-value (*p* < 0.05). In addition, descriptive statistical methods, namely, frequency and cross-tabulation, as well as Microsoft Excel, were also used to describe the basic features and provide simple summaries of the data. 

For the paired *t*-test, there are 4 variables that were computed before and after watching the video in both groups. The variables are the knowledge of (a) mercury and its dangers; (b) the relationship between the Minamata disease and mercury; (c) the effect of mercury on human beings and other living creatures; and (d) the willingness to put a stop to ASGM activities. For the independent samples *t*-test, since this method is designed to inspect the significant differences between the dependent and other independent variables, the dependent variable is the location of the respondents (SR and NR). Minamata-related questions were considered important to ask respondents because mercury is the cause of Minamata disease, which has become a global problem [5,39,40]. In addition, the Indonesian government has made serious efforts to implement the Minamata Convention, which has been ratified since 2017.

### 3.3. Using Video as Transformative Boundary Object (TBO)

#### 3.3.1. Transformative Boundary Object (TBO)

The concept of transformative boundary objects (TBOs) was originally born from the development of boundary objects (BO) to promote collaboration. Star and Griesemer, in 1989 [41], defined a boundary object as an object that facilitates communication among multiple social worlds, and that object is a part of them that has a different identity. The concept of TBOs [42] introduced in this study was developed for the first time by Tsurusaki et al. in 2012 in field education, allowing for a type of hypostatization that not only synchronizes activity and allows the integration of knowledge across the world but also allows for the transformation of the community of the boundary itself. This concept was implemented in a case study by a teacher of 6th-grade students and resulted in evidence that showed that it can help to create a context where students could transform borders [43]. 

The use of video as a type of TBO can be found in a study by Matsumoto in 2018. The video used in his study focused on dementia problems as a means to facilitate the formation of community practices (CoPs) for dementia treatments that cross local boundaries and enact the process of transformative learning [44].

#### 3.3.2. Video to Increase Awareness of Mercury Intoxication

In our study, an approximately-8-min video entitled “The problems in ASGM”, which illustrated three main problems (health, environmental, and social) caused by mercury used in ASGM activities, was used as a TBO to promote knowledge creation and transform rural knowledge with regards to mercury.

The video consisted of a compilation of several documents collected from SRIREP project research activities prior to the COVID-19 outbreak. During the course of the project research, researchers collected not only assessment data but also a variety of digitally documented mediums, such as photographs, audio recordings, and videos. In order to raise awareness of mercury intoxication in ASGM areas, the video for this research study incorporates a number of digital segments, including videos and photographs taken during our project members’ field research studies in Indonesia and Myanmar. The video was a collaboration among all project members and was finalized in June 2022. Following the lifting of travel restrictions, the video was then able to be shown directly to the local people in Gorontalo, Indonesia. 

The SRIREP project has been conducting a number of studies, such as atmospheric Hg contamination, which focuses on the level of atmospheric Hg contamination in the air, water, soil, and living organisms, including trees, as well as clinical surveys focusing on neurological signs and symptoms, respiratory functions, and the analysis of hair samples taken from artisanal and small-scale gold mining (ASGM) communities in Indonesia and Myanmar since 2018 [45].

In addition, SRIREP project’s members have been operating in collaboration with various stakeholders from different public and private organizations including the Japan Association for the United Nations Environment Program (JAU), Kumamoto Gakuen University, Indonesia Medical Association of Gorontalo City, Indonesia Public Health Association, Indonesia National Nurses Association, Ministry of Natural Resources and Environmental Conservation (MONREC), Myanmar, Kyoritsu, Neurology and Rehabilitation Clinic, Japan, Department of Epidemiology, Graduate School of Medicine, Dentistry and Pharmaceutical Sciences, Okayama University, etc., for series of health seminars and workshops designed to increase the awareness of the health problems related to mercury intoxication [46]. 

The evaluation of mercury’s effects on humans is essential, particularly for communities residing in or near ASGM-affected regions. Early diagnosis of mercury toxicity is one of the most crucial monitoring parameters for preventing mercury poisoning effects. 

The stakeholders’ consent was obtained as part of a research agreement, and they have granted permission for the use of their videos in this health awareness educational video to be shown in ASGM communities in collaboration with local key stakeholders during our project researchers’ field trips.

## 4. Results and Discussion

### 4.1. Sociodemographic Profile of Respondents

The summary of the sociodemographics profile of respondents (sample = 105) can be seen in Figure 5a–e, and the detailed profile of respondents within the SR (sample = 64) and NR (sample = 41) groups can be seen in Table 2 as below.

In terms of gender, 85% of respondents were female, whilst 15% were male. The same trend is also shown in each group, where females make up more than 80% of the total respondents. In the present study, we benefited from respondents who were predominantly female because many studies show that females tend to have an important role in family matters such as decision-making [29,30,47,48], as well as family planning [49], especially in promoting and improving family and child health [50]. Regarding marital status, about 80% of the respondents were married, with the same results being shown from both groups.

In terms of age, the average age was 39 years old. The SR group was predominantly composed of respondents in the range of age 31–40 years old (31.3%), while in the NR group, the predominant age range was 41–50 years old (36.3%).

In connection with the educational level of overall respondents, more than 40% of them had at least a high school education level. The percentage of respondents who completed their education at the high school level in the NR group was lower than the SR group, but within the NR group, almost 15% had completed their education at the post-graduate level, while in the SR group, no one had completed their education to this level.

“Housewife” was the most common occupation amongst the respondents (49%), being followed by the occupation of “other” (23%), indicating that such respondents supported their livelihood through irregular work. Although the majority of the respondents were women who worked mostly as housewives and who were not in direct contact with ASGM activities, the role and influence of mothers in children’s education is significant [51,52,53]. Mothers who have early knowledge of the dangers of mercury will be able to transfer this knowledge to their children, thus preventing their children from working in the mines. In addition, female respondents who designated their occupation in the “others” category are assumed to have been doing irregular work around the ASGM site as cooks, laundresses, etc., to meet the needs of the miners. Apart from children, there were also women found working in the mines. Serious health impacts of mining activities for those living in locations around mines, including impaired growth and development of children, with long-term impacts such as the threat of permanent disability, have been found in several studies in Indonesia [1,54,55,56].

For both sample groups, the occupational composition is dominated by “housewife”, followed by “other”, with the main difference between the two being the third most common occupation of each group, namely “farmer” (8%) in the SR group, and “civil servant” (20%) in the NR group. One unique distinction can be seen in the fact that there were no civil servants in the SR group and no farmers in the SR group. Although the data in Table 2 shows that no one worked as a gold miner in either group, mercury can still have an impact not only on miners but also on those living around the mine and even those living off-site. In this sense, the provision of knowledge to those living around the mining site and neighboring districts can be considered valuable as it can prevent them from working in the mine. Based on personal communication with the head of West Tulabolo village, it is known that even those who work as farmers are welcome to supplement their income via work at the mine. Furthermore, miners not only come from within the village; some also come from neighboring districts and even from outside the province [29]. Given this, everyone situated around the mining sites could be suspected of working in the mine due to the overwhelming incentives which mining offers, meaning that the facilitation of early knowledge related to mercury should not be tied to gold miners alone.

### 4.2. Rural Knowledge Transformation in Terms of Mercury

At this stage, the paired samples *t*-test was employed to determine whether there was a notable transformation in rural knowledge regarding mercury before and after watching the presented educational video and, if there was, the overall extent/significance of this transformation. Paired samples *t*-test is suitable for determining whether there is a difference in the means of two groups that are paired or related. In this study, four pairs of questions on a Likert scale were asked, and each individual score was computed before and after watching the video within the two groups used for the analysis. A Likert chart is a scaled chart commonly used in education and social science research for rating respondents’ attitudes on a qualitative level [57]. Three pairs of questions were scored by respondents using the scoring criteria of (1) not at all, (2) a little, (3) more and less, (4) enough, and (5) know well; while the last pair of questions used the score (1) strongly disagree, (2) disagree, (3) doubtful, (4) agree, and (5) strongly agree for the closing question to find out the direction of each respondent’s decision based on the knowledge they had at that time. Logically, the higher the mean score, the more knowledgeable they are in terms of mercury and the stronger their willingness to shut down ASGM activities.

Table 3 presents the result of the paired sample *t*-test before and after watching the video within SR and NR groups. The results show that comparison of the score of mean before and after watching the video, both in the SR and NR groups, has increased in all variables: knowledge of mercury and its dangers in the SR group (before = 2.19, after = 3.11) and NR group (before = 2.15, after = 3.88); knowledge of the relationship between the Minamata disease and mercury in the SR group (before = 1.09, after = 2.44) and NR group (before = 1.63, after = 3.22); knowledge of the effects of mercury on human beings and other living creatures in the SR group (before = 2.13, after = 2.88) and NR group (before = 2.20, after = 3.51); and the willingness to stop the ASGM activities in the SR group (before = 2.69, after = 3.08) and NR group (before = 3.15, after = 3.68).

Results also revealed that there was a statistically significant difference in all variables in the two groups, which was indicated by *p*-value < 0.05: knowledge of mercury and its dangers in the SR group (*p* = 0.000) and NR group (*p* = 0.000); knowledge of the relationship between the Minamata disease and mercury in the SR group (*p* = 0.000) and NR group (*p* = 0.000); knowledge of the effect of mercury on human beings and other living creatures in the SR group (*p* = 0.000) and NR group (*p* = 0.000); and the willingness to stop the ASGM activities in the SR group (*p* = 0.008) and NR group (*p* = 0.004). Statistically, all of the results within the two groups are significant at the 1% level, which means this result reached a 99% confidence level.

Based on this result, it can be concluded that the video, “The problems in ASGM”, is a potentially effective TBO with which to transform rural knowledge in terms of mercury. The effectiveness of video as a medium by which to transform knowledge and skills was also found in the field study of health conducted by Noordman et al. in 2014 [58], which showed the possibility of improving the knowledge of communication skills and motivational interviewing skills; Conceição et al., in 2017 [59], demonstrated video’s capabilities in provisionally increasing knowledge of reproductive health and fertility, also being recommended as a health campaign tool; finally, Nakanakupt and Jaichaun, in 2022 [60], demonstrated that the use of video can enhance the knowledge and skills in conducting delivery.

### 4.3. Difference Factors within SR and NR Groups

Respondents’ knowledge in terms of mercury from each group was assessed both before and after watching the video. This was to inspect whether there were any different factors within the two groups, and whether those differences were statistically significant. Understanding the differences between the two groups is necessary as a basis for composing the initial decision-making steps and analysis framework for each group. Since the data will be analyzed by comparing the mean of unrelated variables from each group, the independent samples *t*-test was employed. The mean score from each group can be read by finding the value assignment in the table.

#### 4.3.1. Difference Factors within the SR and NR Groups before Watching the Video

Table 4 below shows the variable definition and comparable variables before watching the video within the SR and NR groups using the independent samples *t*-test. The test revealed that there were four variables that had statistically significant differences: (1) Have you ever heard about the Minamata disease? (SR mean = 1.91, NR mean = 1.68, *p* = 0.003); (2) Do you know about the Minamata disease? (SR mean = 1.14, NR mean = 1.63, *p* = 0.002); (3) Do you know the relationship between the Minamata disease and mercury? (SR mean = 1.91, NR mean = 1.68, *p* = 0.000); and (4) Do you have the willingness to put a stop to ASGM activities? (SR mean = 2.13, NR mean = 2.20, *p* = 0.034). All variables’ *p*-values were <0.05, with significance at the 1% level, meaning that this result reached a 99% confidence level. For the other three variables, there was a slight difference in each of the mean scores; however, there was no statistically significant difference since the *p*-value was > 0.05.

Respondents in the NR group were more likely to have heard of the Minamata disease (31.7%), and also to know a little about the disease (17.1%), compared to the SR group, wherin about 90% had never heard of Minamata disease and only 10% knew a little about the disease. Similarly, regarding the relationship between mercury and the Minamata disease, in the SR group, 90% did not know about it, compared to the NR group where 30% had little or sufficient knowledge about it and 70% did not know anything about it.

Regarding the willingness to stop ASGM activities, for the SR group, most respondents expressed doubt (45%), followed by disagreement (including strongly disagree at 39%), with only 16% in agreement (including strongly agree). For the NR group, there was an equal portion who were doubtful and agreed (including strongly agree 37%), and 26% disagreed (including strongly disagree).

#### 4.3.2. Difference Factors within the SR and NR Groups after Watching the Video

Table 5 shows the variable definition and comparable variables after watching the video within the SR and NR groups using independent samples *t*-test. The test revealed that there were four variables that had statistically significant differences: (1) After watching the video, do you understand about mercury and its dangers? (SR mean = 3.11, NR mean = 3.88, *p* = 0.003); (2) After watching the video, do you understand the relationship between the Minamata disease and mercury? (SR mean = 2.44, NR mean = 3.22, *p* = 0.002); (3) After watching the video, do you understand mercury’s effect on human beings and other living creatures? (SR mean = 2.88, NR mean = 3.51, *p* = 0.009); and (4) After watching the video, do you agree that ASGM activities should be halted? (SR mean = 3.08, NR mean = 3.68, *p* = 0.014). All variables’ *p*-values were <0.05, and significance was at the 1% level (reaching 99% confidence level). For the other two variables, there was a slight difference in each of the mean scores; however, there was no statistically significant difference since the *p*-value was >0.05.

Respondents in the NR group were more likely to understand mercury and its dangers, with more than 40% having enough of an understanding and almost 35% knowing well compared to the SR group, where the majority of respondents knew a little (42%), only 25% understood enough, and 18% knew well. Similarly, regarding the relationship between mercury and the Minamata disease, the NR group showed a greater understanding compared to the SR group. For the SR group, only 3% knew well, 23% knew enough, and more than 50% knew a little, while for the NR group, 22% knew well, 27% had enough knowledge about it, and only 20% knew a little. 

Furthermore, in their understanding of mercury’s effects on human beings and other living creatures, for the SR group, 50% of the respondents in the group had little understanding, 25% understood enough, and only 10% had a good understanding of mercury’s effects on humans and other living creatures. This is compared to the NR group, where more than a third of the respondents understood enough, almost one-fourth had a good understanding, and only 2% had no understanding that mercury affects humans and other living creatures. Here we can see that although the content of the presented video was the same, their capability to understand the content was different. The NR group tended to obtain a better understanding compared to the SR group. This was influenced by their level of prior knowledge, especially in terms of mercury/Minamata disease (see results of Table 4). This is in line with Cai et al. (2022) [61], who revealed that individuals with high prior knowledge could obtain better factual understanding than those with low prior knowledge. Regarding the cessation of ASGM activities, for the SR group, most respondents (48%) expressed doubt, 28% agreed (including strongly agree), and only 16% disagreed (including strongly disagree); for the NR group, more than 60% expressed agreement (including strongly agree), 17% doubt, and 22% disagreement (including strongly disagree).

### 4.4. The Willingness to Put an End to ASGM Activities

The willingness to end ASGM activities within the two groups can be statistically measured from the mean before and after watching the video. Although NR had more desire and greater agreement towards stopping ASGM activities, both groups experienced a change in decision towards “agreeing” to stop ASGM activities, and both *t*-test results also showed a significant difference at the 1% level. For the SR group, the mean score before was 2.69, which is between “disagree” and approaching “doubtful”, and the mean score after watching the video was 3.08 with a tendency towards “doubtful”. For the NR group, the mean score before was 3.15, which is between “doubtful” and a tendency towards “agree”, and the mean score after watching the video was 3.68, which demonstrates a stronger inclination towards “agree”.

Nevertheless, results also revealed that in the SR group, despite the fac that after watching the video there was an increase of about 7% of respondents who agreed to disengage with ASGM activities, almost 50% of the respondents were still inclined to doubt, and about 40% tended to disagree.

#### Cultural Factor

It is undeniable that the gift of natural resources of gold reserves in Gorontalo can become an economic foundation for the community, especially for those who live in the area. With simple knowledge, they are able to extract gold and make money, which not only allows them to fulfill their daily needs but has also led to many success stories of miners who have realized their dreams and ambitions. However, it has been explained before that mercury has extremely dangerous impacts upon the environment which will produce a negative legacy for the current generation’s children and grandchildren. There is also the negative impact on health, as well as the social impacts further exacerbating the problems caused by ASGM. Even so, there are still those who are hesitant to put a stop to ASGM activities. However, economic factors may not be the only factors which push miners towards continuing to endure the high risk of occupational accidents.

Another factor that may have an influence is the cultural factor, namely, patriarchal culture. In some areas of Indonesia, including Gorontalo, it is still common to find men in control of the household. Many wives are not allowed to work, and they are given domestic responsibilities to take care of housework and children. Wives willingly accept it and give their devotion to the family. All the economic needs of the household are fully borne by the husband. Husbands are busy working, and some even leave the family for extended periods, so they hardly have time to take care of the children. Unfortunately, this patriarchal culture also eventually earned Indonesia the nickname of the “fatherless” nation. Children have a physical father, but many of them have no psychological father [62].

Based on personal communication with one of the miners’ wives (32 y.o), who possesses a baccalaureate degree from a nursing academy, it is known that her husband asked her not to work and to be responsible for the household and look after the children, despite the fact that she has plentiful opportunities for obtaining a job. Since this was the husband’s wish, the wife had to comply. Although for her, being able to work in accordance with the discipline she studied could provide psychological satisfaction, that hope must be buried in order to obey the husband’s request. On the other hand, the husband who made that decision must hold himself to the responsibility of fulfilling all the needs of the household, including his wife’s personal needs. Even though her husband knows the occupational risks at the ASGM site and would live apart from his family (going back every 2–3 weeks), he still chooses to work as a gold miner. The money he brings home is at least 345 USD per 2–3 weeks, many times higher than the minimum wage of Gorontalo Province (200 USD per month) [63].

This factor could be the reason why miners are still unwilling to stop mining, and there could be other invisible factors that also have an influence. Such other factors influencing the willingness to stop involvement with ASGM activities, however, will not be discussed in this paper, as further studies are required. 

## 5. Follow-Up Activities Undertaken by SRIREP Project

### 5.1. Inviting Community to Join the Transdisciplinary Community Practices (TDCOPs) Currently Being Developed by SRIREP

The SRIREP project, together with researchers from Universitas Negeri Gorontalo (UNG) and related stakeholders, encourages local communities to innovate regional potential so that they can create new jobs that become sustainable livelihood alternatives in the future, allowing for a society that does not depend on ASGM, as well as solutions to poverty-related problems through the development of TDCOPs. The transformation of rural people’s knowledge in terms of mercury will form a strong foundation for changing their values towards the environment, so SRIREP welcomes such communities to join the TDCOPs that are currently being developed in Gorontalo. 

TDCOPs are informal formations where stakeholders from different sectors, including academic researchers with a shared interest in a given subject, engage in sharing, learning, and practicing in one group, with regards to scientific knowledge and local knowledge, to solve social problems that are complex and difficult [64]. Several TDCOPs were formed simultaneously by adapting the TDCOP development stages introduced by Wenger et al. in 2002 [65]. Development stages of TDCOPs in Gorontalo can be seen in Figure 6. Beginning with the conducting of a dialogue, as well as gathering basic information and trust-building among participants, TBOs are formulated, eventually leading to the establishment of a group TDCOP. This is followed by the holding of transformative learning seminars on the importance of protecting the environment, which is expected to be able to transform participants’ values and allow for the realization that the environment is their property that must be preserved so that it might be passed on to their children and grandchildren. Furthermore, an action plan is then developed, where training/seminars for improving certain skills are cultivated. This is followed by the formation of “build and strengthen networks” and the implementation of planned pilot activities. 

### 5.2. TDCOPs in Gorontalo Province

At present, a total of six TDCOPs have been active in Gorontalo. These TDCOPs are made up of multisectoral stakeholders both in Japan and Indonesia. Those who live in the same regency as the ASGM area tend to have little interest in the stopping of ASGM activities, which is why the development of TDCOPs has been expanded in Bone Bolango Regency, particularly in Suwawa, the area closest to the ASGM area. The location of TDCOPs can be seen in Figure 7. 

#### 5.2.1. Karawo Research Group

This group focuses on women’s empowerment in continuing Gorontalo’s traditional embroidery, “Karawo”. The aim is to improve the social status of women by (1) helping them build skills through Karawo seminars/training sessions and (2) earning a decent stable income through stable production and sales by expanding market opportunities abroad such as in Japan. As their social status improves, they will contribute significantly to the household economy which may prevent their family members from heading to the ASGM site. This group was consolidated into an entity dubbed “Karawo Light of Life”. 

#### 5.2.2. Natural Fiber Research Group

This group focuses on the utilization of local knowledge concerning the use of natural fiber in daily life which is understood to hold the potential for development into new industries. This includes local methods of making (1) a rope from sugar palm fiber, which can then be developed into a new product of palm fiber nets that are used as slope protection and as a medium for making green curtains/roofs/walls; (2) a mat from pandanus leaves, “Amongo”, which can be re-designed for other purposes such as geotextile and ceiling/wall ornaments. Currently, experiments on the nets’ functions as slope protection and green curtains, as well as investigations into Amongo as a type of geotextile, are being carried out in Japan and Indonesia. This group was consolidated into an entity dubbed “Life from Natural Fibers”.

#### 5.2.3. Sorghum Research Group

This group focuses on accelerating the growth and cultivation of sorghum, a crop that has many health benefits, and which also has the potential to be developed into new industries. Sorghum seeds are consumed as rice or flour that can be used as raw materials for various foods and confectionaries. The leaves and stems are used as animal feed, and the stems of certain varieties can produce liquid sugar suitable for diabetics. Currently, the inclusion demonstration plot area, sorghum utilization promotion, and marketing channels are being pursued, all while research into the maximum utilization of sorghum continues. Experiments related to fattening goats fed with sorghum stems and leaves are being conducted in Indonesia, and in the near future, SRIREP goat breeding will be launched. This group was consolidated into an entity dubbed “Sorghum is a Fortune Source”.

#### 5.2.4. Eco-Tourism Research Group

This group focuses on the environmental conservation and eco-tourism promotion which led to the achievement of UNESCO Gorontalo Geopark. Gorontalo’s long coastline has the potential to be developed for the purposes of tourism, which would allow for alternative sources of livelihood. Most people who live on the coast work as sailors, but when the season does not allow them to go to sea, many of them work in the mines periodically. Currently, there are several beach locations that are being developed into tourism spots, while simultaneously, a waste management program to address the waste problem is being implemented. Besides the beach, other potencies such as hot springs and bird breeding programs are ready to be developed. This group was consolidated into an entity dubbed “Geo-Cafe Gorontalo”.

#### 5.2.5. Mercury Reduction Technology Research Group

This group focuses on technological development towards the reduction of the amount of mercury released by incinerating amalgam. When the mercury amalgam method is used, the device being developed will suppress the release of mercury vapor into the atmosphere. Such a device will be useful for a society who wants to be free from mercury emissions but still depends on ASGM.

#### 5.2.6. Tulabolo Research Group

This group focuses on the Tulabolo area, which consists of three villages in Suwawa Timur subdistrict. This group is the most recent group formed, with the aim that TDCOPs development activities can be more focused on the involvement of miners, villagers, and village officials.

## 6. Limitations

The factors influencing the willingness to end ASGM activities and discussion related to new industries or alternative livelihood creation for the community through the development TDCOPs will be comprise the limitations of this paper.

## 7. Conclusions

The presentation of a video explaining mercury and the problems that occur as a result of ASGM activities had the potential to significantly transform rural knowledge about mercury and increase local willingness to oppose ASGM, creating a strong foundation for changing local values towards the environment. Moreover, the process of encouraging local communities to innovate regional potential in creating sustainable new jobs or alternative livelihoods under the framework of TDCOPs was shown to be an effective vehicle towards achieving this goal. The use of video as a TBO, reinforced by the development of TDCOPs, can therefore be recommended for application throughout ASGM areas in Indonesia, especially in Gorontalo province.

Four factors were identified as causing significant statistical differences within the SR and NR groups. Compared to the SR group, the NR group were more likely to understand mercury and its danger, the relationship between mercury and the Minamata disease, and mercury’s effects on human beings and other living creatures. In addition, the NR group demonstrated more desire and greater consent in putting a stop to ASGM activities. This information is very useful in determining guidance courses and formulating a framework towards mercury reduction strategies.

## Figures and Tables

**Figure 1 ijerph-20-06640-f001:**
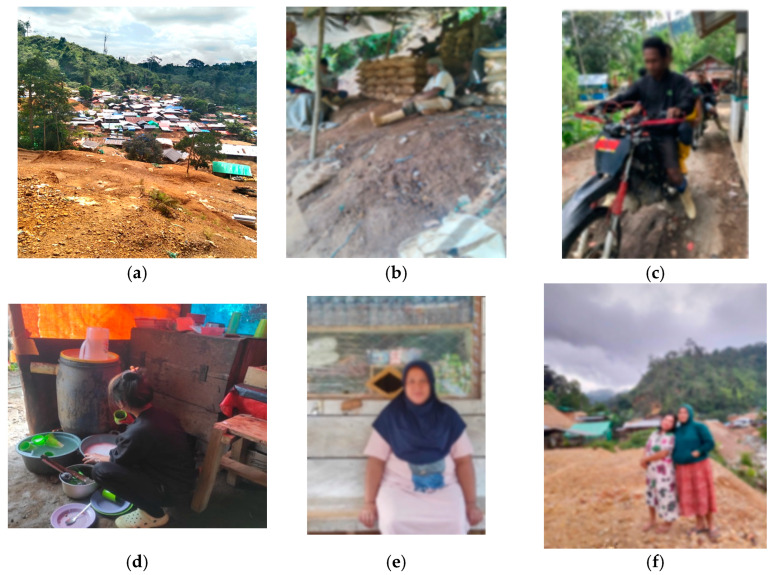
ASGM site in Suwawa Timur subdistrict, Bone Bolango regency, Gorontalo. (**a**) View from above “*Titik Bor* no. 17” resembling a village. (**b**) Miners taking a break in their camp. (**c**) A custom-assembled motorcycle used as public transportation. (**d**) Female food stall owner washing dishes. (**e**) Female small grocery shop owner selling daily needs. (**f**) Women who work as laundresses. Source: Photos of (**a**,**b**,**d**–**f**) were taken by Salma Masuwara and permission was obtained. (**c**) was taken by the first author.

**Figure 2 ijerph-20-06640-f002:**
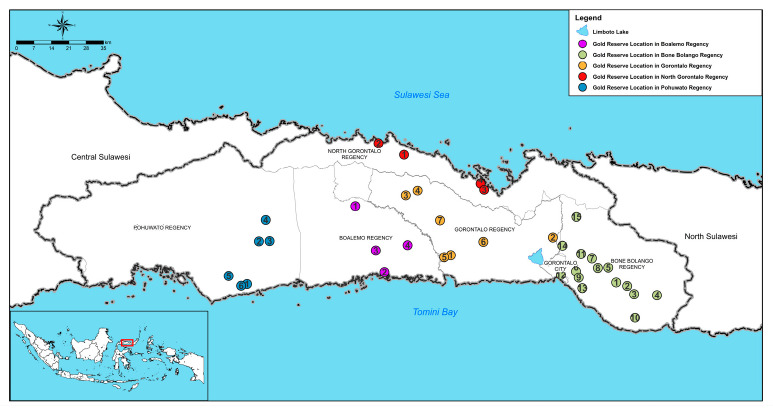
Distribution of gold reserves in Gorontalo province. Note: visualization information from Table 1.

**Figure 3 ijerph-20-06640-f003:**
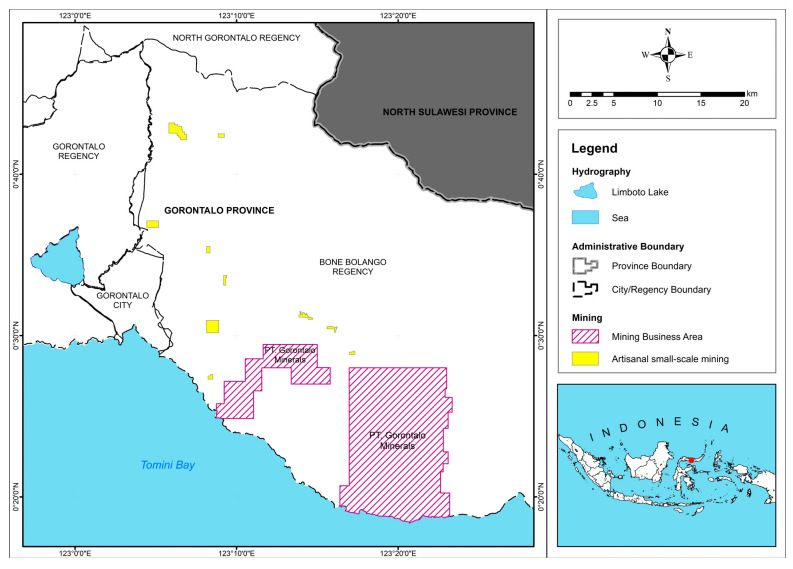
Mining business areas of gold in Bone Bolango Regency, Gorontalo Province.

**Figure 4 ijerph-20-06640-f004:**
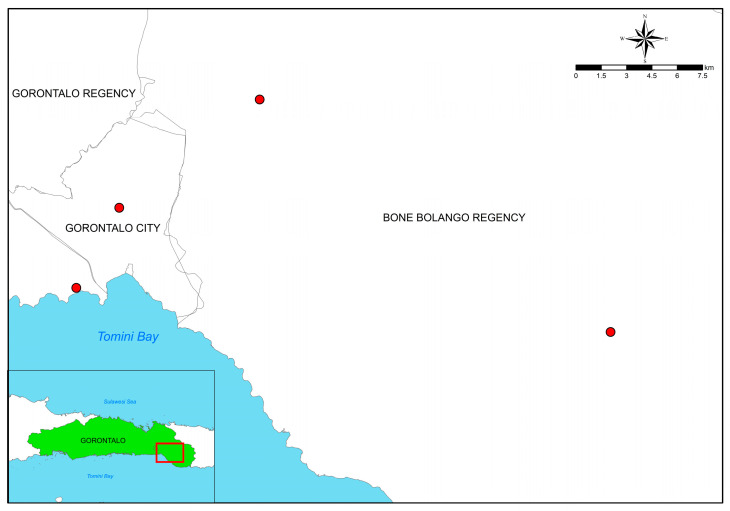
Research sampling location in Gorontalo Province.

**Figure 5 ijerph-20-06640-f005:**
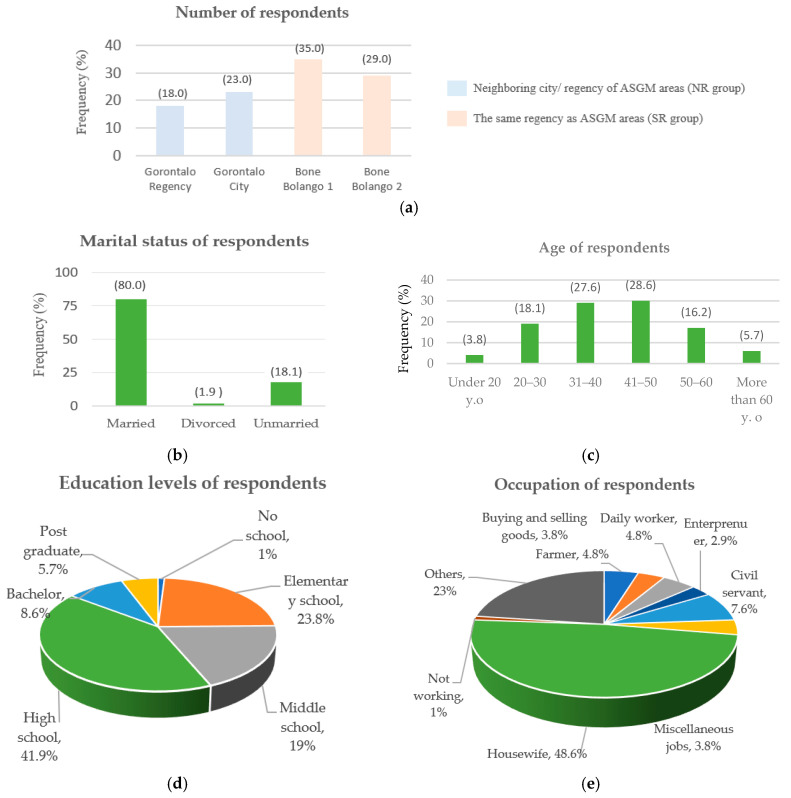
Summary of the sociodemographics profile of respondents: (**a**) study sites and the distribution of respondents on each site; (**b**) marital status of respondents; (**c**) age of respondents; (**d**) education levels of respondents; and (**e**) occupation of respondents.

**Figure 6 ijerph-20-06640-f006:**
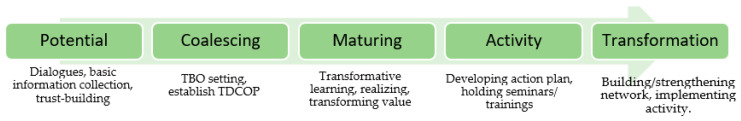
Development stages of TDCOPs in Gorontalo adapted by Wenger et al. in 2002 [65].

**Figure 7 ijerph-20-06640-f007:**
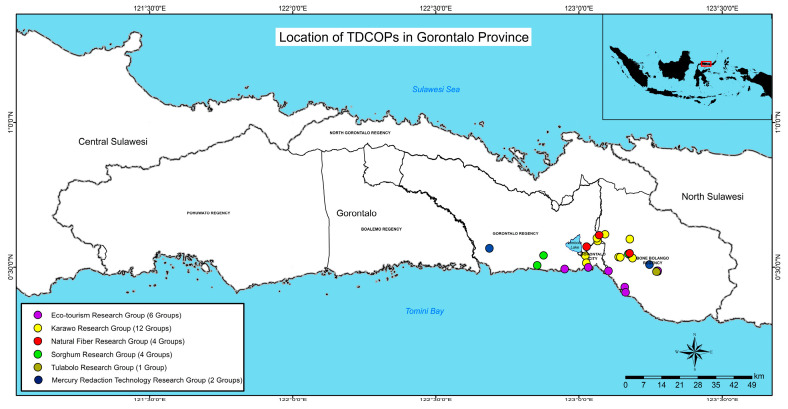
Distribution of TDCOPs development in Gorontalo Province in 2023.

**Table 1 ijerph-20-06640-t001:** Gold reserves in Gorontalo province by region.

Regency	Subdistrict	Village/Area
North Gorontalo	Sumalata	① Hulawa * ② Bulontio *
	Anggrek	③ Ilangata * ④ West Ilangata *
Pohuwato	Buntulia	① Pani mountain ② Ilota Kiri—Hulawa * ③ Ilota Kanan—Hulawa * ④ Baginite mountain—Hulawa * ⑤ Dulumayo ⑥ Buntulia **
Boalemo	Wonosari	① Saritani *
	Dulupi	② Dulupi
		③ Tangga Jaya *
	Paguyaman	④ Saripi **
Gorontalo	Bilato	① Bumela *
	Talaga Biru	② Ulupato Barat *
	Boliyohuto	③ Gunung Boliyohuto **
	Tolanghula	④ Tamaila **
	Asparaga	⑤ Totopo *
	Pulubala	⑥ Pulubala *
	Mootilango	⑦ Pasir Putih—Helumo **
Bone Bolango	Suwawa Timur	① Tulabolo * ② Mak river Tulabolo Timur ③ Motomboto mountain * ④ Kayu Bulan * ⑤ Tilangobula * ⑥ Paduwoma *
	Suwawa	⑦ Huluduotamo *
	Suwawa Tengah	⑧ Tapadaa *
	Suwawa Selatan	⑨ Bulontala Timur *
	Bone Raya	⑩ Tombulilato Kiri *
	Tilong Kabila	⑪ Tunggulo *
	Dumbo Raya	⑫ Talumolo *
	Kabila	⑬ Botutonuo *
	Bulango Utara	⑭ Lomaya *
	Bulango Ulu	⑮ Mongilo Utara *

Note: the designated numbers can be cross-referenced to the village/area numbers specified in Table 1. (*) Locations that have ASGM, (**) ASGM locations that have been closes by the government. Source: Compiled from [17,21,22,23,24,25,30].

**Table 2 ijerph-20-06640-t002:** The sociodemographic profile of respondents within NR and SR groups.

Variables	Number of Respondents	Respondent Group	
		SR	NR
Number of respondents	105 (100.0%)	64 (61.0%)	41 (39.0%)
Gender			
Male	16 (15.2%)	8 (12.5%)	8 (19.5%)
Female	89 (84.8%)	56 (87.5%)	33 (80.5%)
Marital status			
Married	84 (80.0%)	50 (78.1%)	34 (82.9%)
Divorced	2 (1.9%)	0 (0.0%)	2 (4.90%)
Unmarried	19 (18.1%)	14 (21.9%)	5 (12.2%)
Age (years)			
Under 20 years old	4 (3.8%)	4 (6.3%)	0 (0.0%)
20–30	19 (18.1%)	16 (25.0%)	3 (7.30%)
31–40	29 (27.6%)	20 (31.3%)	9 (22.0%)
41–50	30 (28.6%)	15 (23.4%)	15 (36.6%)
50–60	17 (16.2%)	8 (12.5%)	9 (22.0%)
More than 60 years old	6 (5.7%)	1 (1.6%)	5 (12.2%)
Education level			
No school	1 (1.0%)	1 (1.6%)	0 (0.0%)
Elementary school	25 (23.8%)	13 (20.3%)	12 (29.3%)
Middle school	20 (19.0%)	14 (21.9%)	6 (14.6%)
High school	44 (41.9%)	31 (48.4%)	13 (31.7%)
Bachelor	9 (8.6%)	5 (7.8%)	4 (9.80%)
Postgraduate	6 (5.7%)	0 (0.0%)	6 (14.6%)
Occupation			
Farmer	5 (4.8%)	5 (7.8%)	0 (0.0%)
Fishermen	0 (0.0%)	0 (0.0%)	0 (0.0%)
Buying/selling goods	4 (3.8%)	3 (4.7%)	1 (2.4%)
Daily worker	5 (4.8%)	3 (4.7%)	2 (4.9%)
Driver (car/motorcycle)	0 (0.0%)	0 (0.0%)	0 (0.0%)
Entrepreneur	3 (2.9%)	2 (3.1%)	1 (2.4%)
Civil servant	8 (7.6%)	0 (0.0%)	8 (19.5%)
Miscellaneous	4 (3.8%)	4 (6.3%)	0 (0.0%)
Gold miner	0 (0.0%)	0 (0.0%)	0 (0.0%)
Housewife	51 (48.6%)	32 (50.0%)	19 (46.3%)
Not working	1 (1.0%)	0 (0.0%)	1 (2.40%)
Others	24 (22.9%)	15 (23.4%)	9 (22.0%)

**Table 3 ijerph-20-06640-t003:** Paired samples *t*-test before and after watching the video within SR (N = 64, df = 63) and NR (N = 41, df = 40) group.

Variables	Value Assignment		Mean		Std. Dev		*p*-Value ^1^	
			SR	NR	SR	NR	SR	NR
Knowledge of mercury and its dangers		Before After	2.19 3.11	2.15 3.88	1.21 1.27	1.35 1.19	0.000 *	0.000 *
Not at all	=1							
A little	=2							
More and less	=3							
Enough	=4							
Know well	=5							
Knowledge of relationship between the Minamata disease and mercury		Before After	1.09 2.44	1.63 3.22	0.29 1.13	1.11 1.39	0.000 *	0.000 *
Not at all	=1							
A little	=2							
More and less	=3							
Enough	=4							
Know well	=5							
Knowledge of effect of mercury on human beings and other living creatures		Before After	2.13 2.88	2.20 3.51	1.39 1.18	1.36 1.25	0.000 *	0.000 *
Not at all	=1							
A little	=2							
More and less	=3							
Enough	=4							
Know well	=5							
Willingness to put a stop to ASGM activities		Before After	2.69 3.08	3.15 3.68	0.94 1.04	1.24 1.42	0.008 *	0.004 *
Strongly disagree	=1							
Disagree	=2							
Doubtful	=3							
Agree	=4							
Strongly agree	=5							

Note: Based on paired samples *t*-test *p*-value ^1^: * significant at the 1% level.

**Table 4 ijerph-20-06640-t004:** Variable definition and comparable variables before watching the video within the SR and NR groups using independent samples *t*-test.

Variables	Value Assignment	Number of Respondents (Percentage)	Respondent Group (Percentage)		Mean		Std. Dev		*p*-Value ^1^
			SR	NR	SR	NR	SR	NR	
Do you know about mercury?					1.42	1.44	0.50	0.50	0.864
Yes	=1	60 (57.1%)	37 (57.8%)	23 (56.1%)					
No	=2	45 (42.9%)	27 (42.2%)	18 (43.9%)					
Do you know about the dangers of mercury?					2.19	2.15	1.21	1.35	0.871
Not at all	=1	40 (38.1%)	21 (32.8%)	19 (46.3%)					
A little	=2	36 (34.3%)	27 (42.2%)	9 (22.0%)					
More and less	=3	6 (5.7%)	2 (3.1%)	4 (9.8%)					
Enough	=4	17 (16.2%)	11 (17.2%)	6 (14.6%)					
Know well	=5	6 (5.7%)	3 (4.7%)	3 (7.3%)					
Have you ever heard about the Minamata disease?					1.91	1.68	0.29	0.47	0.003 *
Yes	=1	19 (18.1%)	6 (9.4%)	13 (31.7%)					
No	=2	86 (81.9%)	58 (90.6%)	28 (68.3%)					
Do you know about the Minamata disease?					1.14	1.63	0.47	1.13	0.002 *
Not at all	=1	85 (81.0%)	57 (89.1%)	28 (28.3%)					
A little	=2	13 (12.4%)	6(9.4%)	7 (17.1%)					
More and less	=3	0 (0.0%)	0(0.0%)	0 (0.0%)					
Enough	=4	6 (5.7%)	1 (1.6%)	5 (12.2%)					
Know well	=5	1 (1.0%)	0 (0.0%)	1 (2.4%)					
Do you know the relationship between the Minamata disease and mercury?					1.09	1.63	0.29	1.11	0.000 *
Not at all	=1	86 (42.2%)	58 (90.6%)	28 (68.3%)					
A little	=2	12 (11.4%)	6 (9.4%)	6 (14.6%)					
More and less	=3	2 (1.9%)	0 (0.0%)	2 (4.9%)					
Enough	=4	4 (3.8%)	0(0.0%)	4 (9.8%)					
Know well	=5	1 (1.0%)	0 (0.0%)	1 (2.4%)					
Do you think mercury has an effect on human beings and other living creatures?					2.13	2.20	1.39	1.36	0.800
Not at all	=1	52 (49.5%)	33 (51.6%)	19 (46.3%)					
A little	=2	18 (17.1%)	10 (15.6%)	8 (16.5%)					
Doubtful	=3	8 (7.6%)	5 (7.8%)	3 (7.3%)					
Yes	=4	21 (20.0%)	12 (18.8%)	9 (22.0%)					
Absolutely yes	=5	6 (5.7%)	4 (6.3%)	2 (4.9%)					
Do you have a willingness to put a stop to ASGM activities?					2.69	3.15	0.94	1.24	0.034 *
Strongly disagree	=1	12 (11.4%)	7 (10.9%)	5 (12.2%)					
Disagree	=2	24 (22.9%)	18 (28.1%)	6 (14.6%)					
Doubtful	=3	44 (41.9%)	29 (45.3%)	15 (36.6%)					
Agree	=4	16 (15.2%)	8 (12.5%)	8 (19.5%)					
Strongly agree	=5	9 (8.6%)	2 (3.1%)	7 (17.1%)					

Note: Based on independent samples *t*-test *p*-value ^1^: * significant at the 1% level.

**Table 5 ijerph-20-06640-t005:** Variable definition and comparable variables after watching the video within the SR and NR group using independent samples *t*-test.

Variables	Value Assignment	Number of Respondents (Percentage)	Respondent Group (Percentage)		Mean		Std. Dev		*p*-Value ^1^
			SR	NR	SR	NR	SR	NR	
Is the content of the video easy to understand?					1.09	1.07	0.29	0.26	0.716
Yes	=1	96 (91.4%)	58 (90.6%)	38 (92.7%)					
No	=2	9 (8.6%)	6 (9.4%)	3 (7.3%)					
After watching the video, do you understand about mercury and its dangers?					3.11	3.88	1.27	1.19	0.003 *
Not at all	=1	6 (5.7%)	3 (4.7%)	3 (7.3%)					
A little	=2	30 (28.6%)	27 (42.2%)	3 (7.3%)					
More and less	=3	10 (9.5%)	6 (9.4%)	4 (9.8%)					
Enough	=4	33 (31.4%)	16 (25.0%)	17 (41.5%)					
Know well	=5	26 (24.8%)	12 (18.8%)	14 (34.1%)					
After watching the video, do you understand about the Minamata disease and other diseases caused by mercury?					1.17	1.15	0.38	0.36	0.732
Yes	=1	88 (83.8%)	53 (82.8%)	35 (85.4%)					
No	=2	17 (16.2%)	11 (17.2%)	6 (14.6%)					
After watching the video, do you understand the relationship between the Minamata disease and mercury?					2.44	3.22	1.13	1.39	0.002 *
Not at all	=1	17 (16.2%)	11 (17.2%)	6 (14.6%)					
A little	=2	41 (39.0%)	33 (51.6%)	8 (19.5%)					
More and less	=3	10 (9.5%)	3 (4.7%)	7 (17.1%)					
Enough	=4	26 (24.8%)	15 (23.4%)	11 (26.8%)					
Know well	=5	11 (10.5%)	2 (3.1%)	9 (22.0%)					
After watching the video, do you understand mercury’s effect on human beings and other living creatures?					2.88	3.51	1.18	1.25	0.009 *
Not at all	=1	5 (4.8%)	3 (4.7%)	2 (4.9%)					
A little	=2	42 (39.0%)	32 (50.0%)	10 (2.4%)					
More and less	=3	10 (9.5%)	6 (9.4%)	4 (9.8%)					
Enough	=4	31 (29.5%)	16 (25.0%)	15 (36.6%)					
Know well	=5	17 (16.2%)	7 (10.9%)	10 (24.4%)					
After watching the video, do you agree that ASGM activities should be halted?					3.08	3.68	1.04	1.42	0.014 *
Strongly disagree	=1	10 (9.5%)	5 (7.8%)	5 (12.2%)					
Disagree	=2	14 (13.3%)	10 (5.6%)	4 (9.8%)					
Doubtful	=3	38 (36.2%)	31 (48.4%)	7 (17.1%)					
Agree	=4	19 (8.1%)	11 (17.2%)	8 (19.5%)					
Strongly agree	=5	24 (22.9%)	7 (10.9%)	17 (41.5%)					

Note: Based on independent samples *t*-test *p*-value ^1^: * significant at the 1% level.

## Data Availability

The data were generated during the study.

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
