# Peer review of "Rural Knowledge Transformation in Terms of Mercury Used in Artisanal Small-Scale Gold Mining (ASGM)—A Case Study in Gorontalo, Indonesia"

_ijerph, 2023, doi:10.3390/ijerph20176640_

Round 1

Reviewer 1 Report

This manuscript has a major flaw that was interviewing people, mostly females, not envolved in ASGM, before and after seeing a video about mercury, about their willingness to halt ASGM activities, when they do not have the power to do so. The authors mention the importance of women in influencing education and nutrition in families but also recognize the patriarchal social structure of Indonesia and the very limited autonomy of women to determine most aspects of their own existence. This flaw is important in a region were ASGM is the main driver of economy and where ASGM is done by men. Table 2 clearly shows that 0 % of respondents were gold miners. However, if respondents were men, the problem would persist even if they declared willingness to stop ASGM, with or without seeing a video on Hg problems. People lie all the time, not only to Japanese researchers, and tend to say what they think people want to hear. The real issue is if they change their ways, declaring it or not. The study did not address this issue. I would bet high that an hypothetical follow-up would show that the study did not produce any relevant change in ASGM prevalence. 

Other comments

Table 1 and figure 2 bring lots of data unsupported by any reference

Fig. 2 uses different colors for the points on the map but does not inform what these colors stand for.

Line 110-115: authors mention "two primary mining companies" what is meant here by primary? Main?

Line 116-117 " with the remainder of activities being carried out by ASGM" does this means that these companies do not use ASGM?

Fig. 3: The relative position of the shown area and the area in fig.2 is unclear. The figure legend shows a symbol for Limboto Lake but the latter is not shown in the figure. Same for province boundaries.

Line 128-129: fairness and credibility take more than choosing two ASGM and two non-ASGM sites

Line 167: "capture the idea that they are interested". How do you capture an idea? Who does "they" refer to?

Line 177-180: who produced the video and when?

Fig.5: nearly 50% of respondents are housewives and approx. 42% have high school education while most ASGM is done by men with lower education than that.

Line 219: What is the Liekert scale?

Authors use the term rural knowledge but one of the sampling areas is a town.

Questionnaire: Asking about the knowledge about the Minamata disease is very tricky. Japanese know about it, for obvious reasons, but many people have some knowledge on Hg and its effects without necessarily associating it to the Minamata disease.

Line 507: Why was the informed consent statement considered not applicable?

Funding: the location of the Institute for Humanity and Nature (Kyoto) should be mentioned here. 

The language allows some ambiguities, mentioned above

Author Response

Dear Reviewer,

Please kindly see the attachment. Thank you.  

Reviewer 2 Report

The article is a good attempt to explicate the application of an informational video to change some negative practices in the ASGM, particularly the use of Mercury. To do so the researchers carried out a survey with the mix of other instruments to the unmask the potential of these sort of strategies to phased out mercury. After reviewing the piece, I consider that is a condition do significant adjustments to the paper if the authors want to publish.  So, my verdict is reconsidering after major revisions. Below you can see the crucial points and some minor ones. I wish all the best for the authors.

1. Introduction.

This requires to be seriously improved. First, there are some propositions that are unprecise, trying to induce the main features of the ASGM and the use of mercury. Second, when the authors arrive to deal with the strategies towards phased out the use of this heavy metal the background mentioned is weak. So, at the point to deal with the case also the literature regarding policies and their strategies to eliminate the use of mercury in Indonesia is poor analysed. Accordingly, the aim or the argument of the paper seems to be vague.

2. ASGM in Gorontalo

Geographically looks a very interesting descriptive picture. Although the narrative addressing the gold extractive systems in Gorontalo remains neglected. What extractive systems are implemented? Underground, open pit, alluvial or a blend of those? It is important to know it, indeed, the mercury have different bioaccumulation processes according to the production system. Also, what is the amount of persons considered as small and artisanal miners in the regency? And what sort of data we have to measure the quantity of Hg released by ASGM in Gorontalo?

3.2. Material and Methods

It is not clear how the sample was defined, considering that so far, the reader cannot perceive the number of small and artisanal miners that use mercury in the case. I don't understand, so there are two surveys applying to each individual in two different times: (1) before watch the informative video, and (2) after watch the informative video. So, is necessary to indicate the time of the said "before" and "after" to watch the video, either in days, weeks and so on and so forth. Because if the individual is convened to watch the video her or him will be predisposed to respond positively to the stimulus generated by the content, such that, could you argue that the subjects will remember that content and will change its traditions and customs in this extractive sector? Also, could it be interesting to see the video, so a link with that post is an appreciated supplementary data of your research. Finally in mixed methods we require to see the connections between the instruments applied, so the interviews are related to what? I mean, those were applied in the same location and in the same period of time that the surveys? If so, the results will be seriously biased.

4. Results and Discussion

I have my reservations about this section. First of all, the paper does not implement a postpositivist logic, when based on a critical review of the literature about ASGM mercury elimination, the authors can argue and posit hypothesis to be tested in this case. So, the sections is like a kitchen's sink without a thread towards what the paper will novelty add to this interdisciplinary field. Then, I recommend moving forward the analysis oriented to test some propositions rather than entitled some statistical associations without conceptual foundations.

Minor points:

P.1. SRIREP, please mention the mean of this abbreviature.

"TDCOPs" and "TBO" what is the mean of these abbreviatures?

P.1. "Artisanal small-scale gold mining (ASGM) is defined as small gold mining activities that are managed by an individual or small groups/enterprises that generally operate informally without permission/a license, and exploit marginal gold reserves with limited capital investment and production."

1. It seems to me that define the ASGM as informal, illegal, or criminal gold extraction is not necessarily a suitable approach. In many regions of the world these mining systems work in formal or legal conditions, with LAW acknowledgement. The main feature to consider Artisanal/Small mining are the weights of gold extracted. The author's mean is tailored to an illegal or criminal mining viewpoint. Also, one cannot generalise that all the ASGM are move forward by individuals or small mining companies, in Latin America or Africa the communities organise and govern those mining systems. Well, also the production figures posited are very controversial, considering that the ASGM as a whole, in particular cases overpass the production of large scale mining companies.

P.1. "most of which being obtained via ASGM."

2. Here the authors contradict their own logic, stating that a significant amount of Gold extracted in Indonesia became from ASGM.

P. 1. "this is because this method is faster, more effortless, and affordable than other existing methods."

3. I do not agree with the two properties that makes the use of Mercury attractive for the artisanal miners. Considering that technologically other methods are faster and effortless. The main reason to amalgam Au with Hg it is because is available and cheaper.

P. 3. "regencies".

4. If the authors are using American English this term must be adapted to the audience. Regencies, means: "municipality", "county", "city", "town" or "village".

P.4. Table 1 and Figure 2. Please indicate the sources. Those gold reserves are tested/proved, potential or only the delocalisation of the mining fronts? If it so, I mean proved/tested or even potential gold reserves the mining is not artisanal or small at all.

The paper needs to be edited by a native English proofreader.

Author Response

Dear Reviewer

Please kindly see the attachment.

Thank you.

Round 2

Reviewer 1 Report

No comments